# G-Designer: Architecting Multi-agent Communication Topologies via Graph Neural Networks

Guibin Zhang [* 1 2]  Yanwei Yue [* 1]  Xiangguo Sun [* 3]  Guancheng Wan [4]  Miao Yu [5]
Junfeng Fang [2]  Kun Wang [6]  Tianlong Chen [7]  Dawei Cheng [1]

## Abstract

Recent advancements in large language model (LLM)-based agents have demonstrated that collective intelligence can significantly surpass the capabilities of individual agents, primarily due to well-crafted inter-agent communication topologies. Despite the diverse and high-performing designs available, practitioners often face confusion when selecting the most effective pipeline for their specific task: *Which topology is the best choice for my task, avoiding unnecessary communication token overhead while ensuring high-quality solution?* In response to this dilemma, we introduce `G-Designer`, an adaptive, efficient, and robust solution for multi-agent deployment, which dynamically designs task-aware, customized communication topologies. Specifically, `G-Designer` models the multi-agent system as a multi-agent network, leveraging a variational graph auto-encoder to encode both the nodes (agents) and a task-specific virtual node, and decodes a task-adaptive and high-performing communication topology. Extensive experiments on six benchmarks showcase that `G-Designer` is: **(1) high-performing**, achieving superior results on MMLU with accuracy at $84.50\%$ and on HumanEval with pass@1 at $89.90\%$; **(2) task-adaptive**, architecting communication protocols tailored to task difficulty, reducing token consumption by up to $95.33\%$ on HumanEval; and **(3) adversarially robust**, defending against agent adversarial attacks with merely $0.3\%$ accuracy drop. The code is available at https://github.com/yanweiyue/GDesigner.

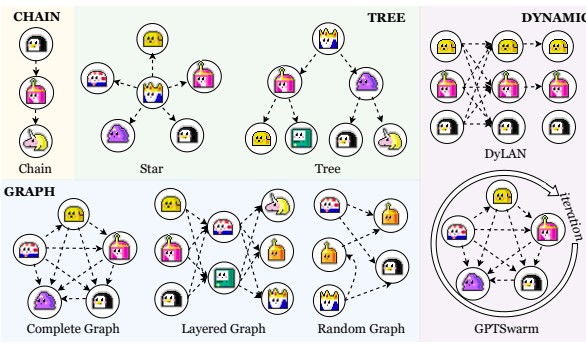

*Figure 1.* Existing practices for LLM-based multi-agent communication topology design.

## 1. Introduction

An LLM-based agent, which integrates the language generation capabilities of LLMs with decision-making and action-execution functionalities (Richards & et al., 2023; Nakajima, 2023; Reworkd, 2023), has exhibited impressive performance across a wide range of tasks, from reasoning (Yao et al., 2023b) and code generation (Shinn et al., 2023) to even more complex applications like video gaming (Wang et al., 2023) and autonomous driving (Jin et al., 2023). Even more exciting, researchers have discovered that combining multiple LLM-based agents–whether implicitly or explicitly–into a team can outperform individual agents when tackling complex tasks (Du et al., 2023; Liang et al., 2023; Wang et al., 2023b; Jiang et al., 2023; Shinn et al., 2023; Zheng et al., 2023; Wu et al., 2023), demonstrating a form of collaborative intelligence reminiscent of human teamwork in multi-agent systems (Zhang et al., 2023b). This emergence of human-esque collective intelligence is fundamentally driven by the design of their topology, *i.e.*, how multi-agents are *connected*, and how they *transmit*, *exchange*, and *assimilate* information reciprocally.

In practice, prior research has extensively explored how multiple instances of LLMs, referred to as agents (Wang et al., 2024; Xi et al., 2023; Gao et al., 2023; Cheng et al., 2024), should be structured and organized to converse, collaborate, debate, or even compete. Various topological designs have been investigated, such as chain (Wei et al., 2022; Hong et al., 2023), tree (Yao et al., 2023a; Wu

---

[*]Equal contribution  [1]Tongji University  [2]NUS  [3]CUHK  [4]UCLA  [5]USTC  [6]NTU  [7]UNC-Chapel Hill. Correspondence to: Kun Wang <wang.kun@ntu.edu.sg>, Dawei Cheng <dcheng@tongji.edu.cn>.

*Proceedings of the 42nd International Conference on Machine Learning*, Vancouver, Canada. PMLR 267, 2025. Copyright 2025 by the author(s).

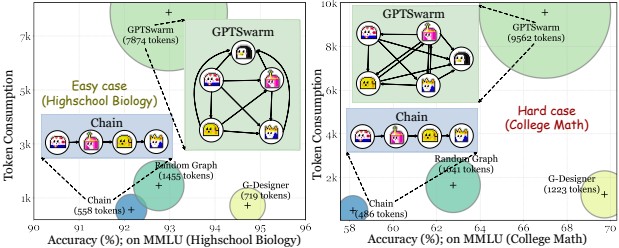

*Figure 2.* The token consumption and accuracy of different multi-agent protocols on two subsets of MMLU dataset, "Highschool Biology" and "College Math", tested with four `gpt-4`-based agents.

et al., 2023), star (Wu et al., 2023), complete graphs (Qian et al., 2024), random graphs (Qian et al., 2024), optimizable graphs (Zhuge et al., 2024; Zhang et al., 2024), and LLM-based networks (Hao et al., 2023; Liu et al., 2023). These elaborately designed communication topologies have demonstrated remarkable performance with minimal human supervision, bridging the gap between individual and collective intelligence. Faced with numerous structures available, an inquisitive practitioner might ask: *how should I select or design a topology that best suits my task at hand?*

The question posed above is *non-trivial* and, at times, *perplexing*. A piece of experimental evidence is presented in Figure 2, where we evaluated the performance of different multi-agent structures on the MMLU dataset (Hendrycks et al., 2021), a collection of multiple-choice questions across various subjects. The results reveal that even within the same dataset, the suitability of different communication topologies varies. ❶ **Simpler Case**: in the simpler "High School Biology" subset, the chain structure performs comparably to the complex GPTSwarm, while consuming significantly fewer tokens (0.5k versus 7.8k). In this case, the chain structure is clearly a more economical choice. ❷ **Harder Case**: However, for the more challenging "College Mathematics" subset, GPTSwarm outperforms the chain structure by 8.75%, primarily attributed to its intricate topology and prompt optimization. In summary, practitioners often find it challenging to *effortlessly identify the most efficient and complexity-adaptive multi-agent topology for a given task.*

In light of this dilemma, we propose the **LLM-based Multi-agent Communication Protocol (MACP)**, establishing standardized guidance for LLM-MA topology design:

> **Multi-agent Communication Protocol (MACP)**: *Given a task/query q, an optimal LLM-MA communication topology for q should satisfy the following protocol logics: (1) Effectiveness: The communication structure must effectively produce the qualified solution for q; (2) Complexity-adaptiveness: The topology should dynamically adjust to the complexity of the task, minimizing communication overhead; (3) Adversarial robustness: The topology should maintain reliable under adversarial attacks.*

The formal definition of MACP is provided in Section 3.3.

To design a communication topology that ideally adheres to the MACP principles, we propose *an effective, adaptive, and robust LLM-powered multi-agent communication graph designer*, termed `G-Designer`. Technically, `G-Designer` first architects a multi-agent graph, where each agent, along with its specific properties (e.g., profile (Li et al., 2023a), external API tools (Zhuang et al., 2023), or knowledge base (Chen et al., 2024a)), is represented as a node, and communication between agents forms the edges. `G-Designer` employs a variational graph auto-encoder to encode the nodes (agents) along with task-specific information, and to decode the resulting collaboration network between agents. This input-dependent paradigm allows `G-Designer` to design **task-adaptive, high-performing communication topology**, which is, at the same time, assured of efficiency and robustness with sparsity regularization. Unlike previous LLM-based multi-agent topology designs, which rely on a static structure for all queries/tasks, `G-Designer` adaptively crafts customized topologies for different domains and tasks, serving as a fully autonomous and flexible assistant for multi-agent system establishment and deployments.

Our contribution can be summarized as follows:

❶ *Protocol Proposal.* We propose the *first* communication protocol tailored for LLM-powered multi-agent systems, MACP, which comprehensively regulates multi-agent topology design across three dimensions: *performance, adaptability, and robustness*, and incisively highlights the shortcomings of existing designs.

❷ *Practical Solution.* We present `G-Designer`, an effective, adaptive, and robust designer of LLM-powered multi-agent communication graphs. By leveraging a variational graph auto-encoder to construct and process the multi-agent network, `G-Designer` decodes task-adaptive and high-performing agent communication, which is also equipped with strong robustness against agent-rooted adversarial attacks via dynamic topology adjustment.

❸ *Experimental Validation.* Extensive experiments across six benchmarks show that `G-Designer` is: **(1) high-performing**, surpassing state-of-the-art topologies by $0.20\% \sim 4.10\%$ on MMLU and HumanEval; **(2) task-adaptive**, dynamically adjusting topology complexity with task awareness, outperforming state-of-the-art methods on MMLU with a cost of merely $1.5e+5$ compared to their $2.6e+6$, reducing token consumption by up to $92.24\%$; and **(3) adversarially robust**, defending against agent adversarial attacks with merely $0.3\%$ accuracy drop.

## 2. Related Works

**LLM-agent Collaboration** Recent research has explored various multi-agent communication topologies, including: **(1) Non-interactive**, where agents operate independently without inter-agent communication, as employed in systems like LATM (Zhang et al., 2023a) and LLM-Debate (Du

et al., 2023); **(2) Chain**, where agents are arranged in a sequential structure, each receiving the output from its predecessor and passing information to its successor, utilized by ChatDev (Qian et al., 2023), MetaGPT (Hong et al., 2023), and L2MAC (Holt et al., 2024); **(3) Star**, where a central administrative agent (often referred to as a commander, teacher, *etc*.) directs subordinate agents, seen in AutoGen (Wu et al., 2023), SecurityBot (Yan et al., 2024), and MiniGrid (Zhou et al., 2023); **(4) Tree**, where a root agent hierarchically manages multiple child agents, as in SoA (Ishibashi & Nishimura, 2024); and **(5) Graph**, encompassing complete graphs (Qian et al., 2024; Zhuge et al., 2024) and random graphs (Qian et al., 2024), among others.

**Multi-agents as Graphs** Graphs, as a fundamental data structure for organizing and representing relationships between entities (Zhang & Chartrand, 2006), are widely adopted in the pre-LLM era as a powerful tool to facilitate effective communication in multi-agent reinforcement learning (MARL) (Pesce & Montana, 2023; Hu et al., 2024; Liu et al., 2022). With the rise of LLMs and the proliferation of LLM-based agents (Chen et al., 2023a; Cohen et al., 2023; Hua et al., 2023), researchers have similarly recognized that interactions among multiple agents can naturally be modeled from a graph-based perspective (Chen et al., 2023b; Zhuge et al., 2024; Qian et al., 2024; Liu et al., 2023). Early attempts are implicit, like ChatEval (Chan et al., 2023), AutoGen (Wu et al., 2023), and DSPy (Khattab et al., 2023). More recent practices including ChatLLM (Hao et al., 2023), DyLAN (Liu et al., 2023), GPTSwarm (Zhuge et al., 2024), and MacNet (Qian et al., 2024), have explicitly represented the organization of multiple agents as a graph. However, all these attempts, whether predefined or iteratively optimized, remain *input-independent*. Consequently, they fail to be task-aware and adaptively design topologies that suit the complexity of the specific task.

# 3. Formalization

This section establishes the notation, formalizes key concepts from a topology perspective, and formally defines our proposed multi-agent communication protocol.

## 3.1. Topology Structure

We model the multi-agent system as a directed graph $\mathcal{G} = (\mathcal{V}, \mathcal{E})$, where $\mathcal{V} = \{v_1, \ldots, v_N\}$ represents the set of nodes (with $N = |\mathcal{V}|$) and $\mathcal{E}$ denotes the set of edges. Each node $v_i \in \mathcal{V}$ corresponds to an agent, formalized as:

$$v_i = \{\texttt{Base}_i, \texttt{Role}_i, \texttt{State}_i, \texttt{Plugin}_i\}, \quad (1)$$

where each agent $v_i$ is composed of four key elements: (1) $\texttt{Base}_i$, the language model instance powering $v_i$; (2) $\texttt{Role}_i$, the agent's pre-assigned role or function; (3) $\texttt{State}_i$, representing the agent's accumulated knowledge and interaction history; and (4) $\texttt{Plugin}_i$, a set of external

tools or plugins available to $v_i$, such as web searchers (Ma et al., 2023), code compilers (Richards & et al., 2023; Wu et al., 2023; Hong et al., 2023; Bouzenia et al., 2024; Ishibashi & Nishimura, 2024), or file readers (Zhuge et al., 2024; Richards & et al., 2023). Each LLM-based agent $v_i$ receives prompt $\mathcal{P}$ and generates response $\mathcal{R}_i$:

$$\mathcal{R}_i = v_i(\mathcal{P}) = v_i(\mathcal{P}_{\text{sys}}, \mathcal{P}_{\text{usr}}), \quad (2)$$

where $\mathcal{P}_{\text{sys}} = \{\texttt{Role}_i, \texttt{State}_i\}$ represents the system prompt encompassing its role and state, and $\mathcal{P}_{\text{usr}}$ denotes the user prompt, which possibly includes the given tasks, responses/instructions from other agents and externally retrieved knowledge.

The connectivity of $\mathcal{G}$ can also be characterized by a (non-symmetric) adjacency matrix $\mathbf{A} \in \{0, 1\}^{N \times N}$, where $\mathbf{A}[i, j] = 1$ if $e_{ij} = (v_i, v_j) \in \mathcal{E}$, otherwise 0. Each edge $e_{ij} \in \mathcal{E}$ represents the flow of information from $v_i$ to $v_j$.

## 3.2. Communication Pipeline

Given a query/problem $\mathcal{Q}$, the multi-agent system engages in $K$ rounds of interactive utterances, which collaboratively drive the agents toward producing the final solution $a^{(K)}$ based on their cumulative dialogue exchanges. At the beginning of the $t$-th dialogue round, a mapping function $\phi$ is applied to determine the execution index for each agent:

$$\phi: \mathcal{G} \longmapsto \sigma, \ \sigma = [v_{\sigma_1}, v_{\sigma_2}, \cdots, v_{\sigma_N}], \\ \text{s.t.} \ \forall i > j, \ \ v_{\sigma_i} \notin \mathcal{N}_{\text{in}}(v_{\sigma_j}), \quad (3)$$

where $\sigma$ is the execution sequence of agents, $\mathcal{N}_{\text{in}}(v_{\sigma(j)})$ denotes the in-neighborhood of $v_{\sigma(j)}$, and the constraint ensures that an agent $v_{\sigma(i)}$ can only execute after any agent $v_{\sigma(j)}$ from which it receives information. Once the execution order is determined, each agent proceeds to perform input-output operations sequentially:

$$\mathcal{R}_i^{(t)} = v_i(\mathcal{P}_{\text{sys}}^{(t)}, \mathcal{P}_{\text{usr}}^{(t)}), \ \mathcal{P}_{\text{usr}}^{(t)} = \{\mathcal{Q}, \cup_{v_j \in \mathcal{N}_{\text{in}}(v_i)} \mathcal{R}_j^{(t)}\} \quad (4)$$

where $\mathcal{R}_i^{(t)}$ represents the output of $v_i$, which could be a rationale, an answer, or a partial solution, depending on the specific context. The output $\mathcal{R}_i^{(t)}$ is generated based on the system prompt $\mathcal{P}_{\text{sys}}^{(t)}$ and the context prompt, consisting of the query $\mathcal{Q}$ and messages from other agents. At the end of each dialogue, an aggregation function is adopted to generate the answer/solution $a^{(t)}$:

$$a^{(t)} \leftarrow \text{Aggregate}(\mathcal{R}_1^{(t)}, \mathcal{R}_2^{(t)}, \cdots, \mathcal{R}_N^{(t)}). \quad (5)$$

The implementation of the Aggregate function is flexible, with possible options including majority voting (Chen et al., 2024b; Zhuge et al., 2024; Li et al., 2024), aggregating all agents' responses and delegating one agent to provide the final answer (Wu et al., 2023; Jiang et al., 2023; Liu et al., 2023; Zhang et al., 2024), or simply using the output of

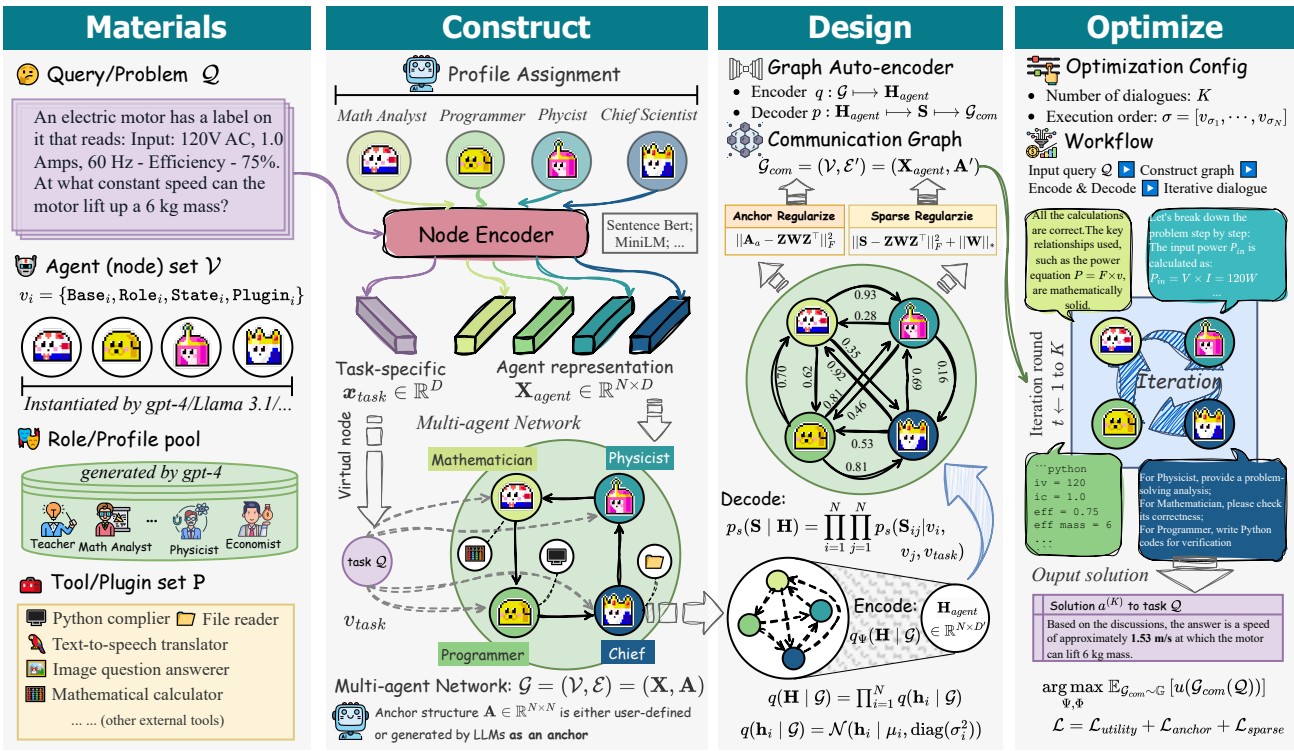

Figure 3. The designing workflow of our proposed `G-Designer`.

the last agent $\mathcal{R}_{\sigma_N}^{(t)}$ (Qian et al., 2024). Through $K$ rounds of utterances, either predefined (Qian et al., 2024) or determined by an early-stopping mechanism (Liu et al., 2023), the overall system $\mathcal{G}$ produces the final answer $a^{(K)}$ for $\mathcal{Q}$.

### 3.3. MACP Protocol

We give the formal definition of MACP Protocol as follows:

**Definition 3.1 (Multi-agent Communication Protocol).** Given an LLM-based multi-agent system $\mathcal{G} = (\mathcal{V}, \mathcal{E})$, we establish the following objective as optimization principle:

$$\min_{\mathcal{G} \in \mathbb{G}} \Big[ -u\big(\mathcal{G}(\mathcal{Q})\big) + \beta_1 \cdot ||\mathcal{G}|| + \beta_2 \cdot \big|\hat{\mathcal{G}}(\hat{\mathcal{Q}}) - \mathcal{G}(\mathcal{Q})\big| \Big], \quad (6)$$

where $\mathbb{G}$ represents the feasible parameter space of $\mathcal{G}$, $u(\cdot)$ is the utility evaluator, $||\mathcal{G}||$ measures the computational and communication overhead of the entire graph, and $\hat{\mathcal{Q}}$ and $\hat{\mathcal{G}}$ denote the query description and the multi-agent system after adversarial perturbation, respectively. The first term in Equation (6) corresponds to **high performance**, aiming to maximize the utility of the system's output; the second term addresses **task-adaptiveness**, seeking to minimize system complexity to reduce power consumption and economic cost; and the third term focuses on **robustness**, constraining the deviation of system output under adversarial attacks.

### 4. G-Designer

Figure 3 illustrates how `G-Designer` adaptively designs communication topologies for any given query. Specifi-

cally, the process begins with a few "raw materials": the input query $\mathcal{Q}$, the agent set $\mathcal{V}$, the profile pool, and the toolset. In the *Construct* stage, `G-Designer` leverages a node encoder to construct a multi-agent network along with a task-specific virtual node. In the *Design* stage, a graph auto-encoder is employed to decode the communication graph topology $\mathcal{G}_{com}$, which is leveraged for multi-round inter-agent collaboration in the *Optimize* stage.

### 4.1. Multi-agent Network Construction

Given an input query $\mathcal{Q}$ and a set of LLM-agents $\mathcal{V}$, `G-Designer` aims to design a task-adaptive and effective communication topology $\mathcal{G}_{com}$. We begin by assigning each agent a unique role and profile, as previous research (Wang et al., 2023b) has shown that assigning distinct personas or roles to LLM-based agents can enhance cognitive synergy. Based on these roles, different external tools are allocated to the agents (*e.g.*, Mathematica for a math analyst, Python compiler for a programmer). Thus, we successfully initialize each agent $v_i$ as $\{\texttt{Base}_i, \texttt{Role}_i, \texttt{State}_i, \texttt{Plugin}_i\}$, as defined in Equation (1).

We proceed to construct a structured multi-agent network as input to `G-Designer`, represented as $\mathcal{G} = (\mathbf{X}_{agent}, \mathbf{A})$, where $\mathbf{X}_{agent} \in \mathbb{R}^{N \times D}$ is the node (agent) feature matrix and $\mathbf{A} \in \mathbb{R}^{N \times N}$ represents the connectivity matrix. For the feature matrix $\mathbf{X}_{agent}$, we employ a node encoder to transform each agent's unique profile into a fixed-length

embedding representation:

$$\mathbf{x}_i \leftarrow \text{NodeEncoder}\left(\mathcal{T}(\text{Base}_i), \text{Role}_i, \mathcal{T}(\text{Plugin}_i)\right), \quad (7)$$

where $\mathcal{T}(\cdot)$ extracts the textual description of the agent's LLM backbone and its assigned plugins, and $\text{NodeEncoder}$ can be realized using small and lightweight text embedding models (Reimers, 2019). After encoding the individual agents, we aim to ensure that the multi-agent network incorporates information related to the query $\mathcal{Q}$, as this query-dependent approach enables G-Designer to be task-aware and adaptive. To this end, we introduce an additional *task-specific virtual global node* $v_{task}$, which is bidirectionally connected to all agent nodes, enabling a global "storage sink" and facilitating smoother information flow among agents (Shirzad et al., 2023; Tan et al., 2023; Rosenbluth et al., 2024). This task node is encoded by the NodeEncoder as follows: $\mathbf{x}_{task} \leftarrow \text{NodeEncoder}(\mathcal{Q})$.

After obtaining the agent node features $\mathbf{X}_{agent} = [\mathbf{x}_1, \mathbf{x}_2, \ldots, \mathbf{x}_N]^\top$ and the task-specific embedding $\mathbf{x}_{task}$, we provide a simple *anchor topology* $\mathbf{A}_{anchor} \in \{0,1\}^{N \times N}$, which serves as a starting point for G-Designer's topology design process. For instance, given a code generation task with three agents: manager/programmer/code reviewer, the anchor topology could be configured as a chain structure, *i.e.*, "manager → programmer → reviewer", reflecting the typical workflow of code completion. The anchor topology, being either user-defined or automatically generated by LLMs, is often simple and sub-optimal[1]. However, it provides a foundational reference and prior knowledge for G-Designer's subsequent optimization process. We incorporate the task-specific vertex $v_{task}$ and its corresponding edges and obtain $\tilde{\mathbf{A}}_{anchor} \in \{0,1\}^{(N+1) \times (N+1)}$. Consequently, we establish a task-specific multi-agent network $\tilde{\mathcal{G}}$:

$$\begin{aligned} \tilde{\mathcal{G}} &= \left(\begin{bmatrix} \mathbf{X}_{agent} \\ \mathbf{x}_{task}^\top \end{bmatrix}, \tilde{\mathbf{A}}_{anchor}\right) = (\tilde{\mathcal{V}}, \tilde{\mathcal{E}}) \\ &= \left(\mathcal{V} \cup \{v_{task}\}, \mathcal{E} \cup \{\overset{\longleftrightarrow}{(v_i, v_{task})} | v_i \in \mathcal{V}\}\right), \end{aligned} \quad (8)$$

where $\begin{bmatrix} \mathbf{X}_{agent} \\ \mathbf{x}_{task}^\top \end{bmatrix}$ can also be jointly denoted as $\tilde{\mathbf{X}}$.

### 4.2. Designing Communication Topology

Building upon the task-specific multi-agent network $\tilde{\mathcal{G}}$, G-Designer seeks to establish a more fine-grained and precise communication topology $\mathcal{G}_{com}$. Drawing inspiration from the variational graph auto-encoder (VGAE) framework (Kipf & Welling, 2016; Zhao & Zhang, 2024), G-Designer employs a VGAE-based encoder-decoder $f_v$ to generate the multi-agent interaction topology:

$$\mathcal{G}_{com} = f_v(\tilde{\mathcal{G}}; \Theta_v) = p(\mathcal{G}_{com} \mid \mathbf{H}) q(\mathbf{H} \mid \tilde{\mathbf{X}}, \tilde{\mathbf{A}}_{anchor}), \quad (9)$$

[1]We discuss the substantial performance improvement of G-Designer over the anchor topology in Section 5.4.

where $f_v$ is the encoder-decoder architecture with parameters $\Theta_v$, $q(\cdot)$ is the encoder module, $p(\cdot)$ is the decoder module. The encoder utilizes posterior probabilities to encode the node embeddings into low-dimensional latent vector representations $\mathbf{H}_{agent}$, which can be formulated as:

$$\begin{aligned} q(\mathbf{H}_{agent} \mid \tilde{\mathbf{X}}, \tilde{\mathbf{A}}_{anchor}) &= \prod_{i=1}^{N} q(\mathbf{h}_i \mid \tilde{\mathbf{X}}, \tilde{\mathbf{A}}_{anchor}), \\ q(\mathbf{h}_i \mid \tilde{\mathbf{X}}, \tilde{\mathbf{A}}_{anchor}) &= \mathcal{N}(\mathbf{h}_i \mid \boldsymbol{\mu}_i, \text{diag}(\boldsymbol{\sigma}_i^2)), \end{aligned} \quad (10)$$

where $\mu = \text{GNN}_\mu(\tilde{\mathbf{X}}, \tilde{\mathbf{A}}_{anchor}; \Theta_\mu)$ is the matrix of mean vectors $\mu_i$; similarly $\log(\sigma) = \text{GNN}_\sigma(\tilde{\mathbf{X}}, \tilde{\mathbf{A}}_{anchor}; \Theta_\sigma)$. The choice of GNN backbone can be customized as needed; here, we utilize a simple two-layer GCN (Kipf & Welling, 2017). $\mathbf{h}_i$, $\boldsymbol{\mu}_i$, and $\boldsymbol{\sigma}_i$ denote the $i$-th column of $\mathbf{H}$, $\boldsymbol{\mu}$, and $\boldsymbol{\sigma}$, respectively. The encoder $q(\cdot)$ is parameterized by $\Theta_e = \{\Theta_\mu, \Theta_\sigma\}$. Following the encoding phase, the decoder employs the latent representations to generate a comprehensive blueprint for multi-agent communication. More specifically, the decoder $q(\cdot) = q_c \circ q_s$ first constructs a parameterized, sketched graph $\mathbf{S}$, which is then refined into the final multi-agent communication topology:

$$p(\mathcal{G}_{com} \mid \mathbf{H}_{agent}) = \int_{\mathbf{S}} p_c(\mathcal{G}_{com} \mid \mathbf{S}) p_s(\mathbf{S} \mid \mathbf{H}_{agent}) \, d\mathbf{S}. \quad (11)$$

At the first step, $p_s(\cdot)$ constructs the sketched adjacency matrix $\mathbf{S}$ from the latent representation $\mathbf{H}_{agent}$:

$$p_s(\mathbf{S} \mid \mathbf{H}_{agent}) = \prod_{i=1}^{N} \prod_{j=1}^{N} p_s(\mathbf{S}_{ij} \mid \mathbf{h}_i, \mathbf{h}_j, \mathbf{h}_{task}; \Theta_d), \quad (12)$$

whose detailed derivation is as follows:

$$\begin{aligned} p_s(\mathbf{S}_{ij} = 1 \mid \mathbf{h}_i, \mathbf{h}_j, \mathbf{h}_{task}) &= g(\mathbf{h}_i, \mathbf{h}_j, \mathbf{h}_{task}), \\ &= \text{Sigmoid}((\log(\epsilon) - \log(1 - \epsilon) + \varpi_{ij})/\tau), \end{aligned} \quad (13)$$

where $\varpi = \text{FFN}_d([\mathbf{h}_i, \mathbf{h}_j, \mathbf{h}_{task}])$ with $\text{FFN}_d$ parameterized by $\Theta_d$, $\epsilon \sim \text{Uniform}(0,1)$, and $\tau$ denotes the temperature coefficient. When $\tau$ approaches zero, Equation (13) essentially return the Bernouli sampling result for $\mathbf{S}_{ij}$. The resulting matrix $\mathbf{S} \in [0,1]^{N \times N}$ represents a densely-connected, non-negative graph distribution, indicating an overly complex and resource-intensive pair-wise communication structure, which is not yet suitable for guiding multi-agent collaboration. To align with G-Designer's objectives of task adaptiveness and minimizing costs, we apply a refinement decoder $p_c(\cdot)$ to refine the sketched $\mathbf{S}$ into a compact, sparse, and highly informative communication graph, instantiated by a regularization objective:

$$\begin{aligned} p_c : \underset{\tilde{\mathbf{S}} \in \mathbb{S}}{\arg\max} \; &1/2 \|\mathbf{S} - \mathbf{Z}\mathbf{W}\mathbf{Z}^\top\|_F^2 + \zeta \|\mathbf{W}\|_* + \\ &1/2 \|\mathbf{A}_{anchor} - \mathbf{Z}\mathbf{W}\mathbf{Z}^\top\|_F^2, \; \text{s.t.} \; \tilde{\mathbf{S}} = \mathbf{Z}\mathbf{W}\mathbf{Z}^\top, \end{aligned} \quad (14)$$

where $\mathbf{Z} \in \mathbb{R}^{N \times r}$ is the top-$r$ columns of left singular matrix $\mathbf{S}$, $\zeta$ is a coefficient hyperparameter, $\mathbf{W} \in \mathbb{R}^{r \times r}$ is an optimizable weight matrix, $|| \cdot ||_F$ denotes the Frobenius norm and $||\mathbf{W}||_* = \sum_i \lambda_i$ where $\lambda_i$ is the $i$-th singular value of $\mathbf{W}$. $\tilde{\mathbf{S}} \in \mathbb{R}^{N \times N}$ is the desired sparse topology, which is decomposed as $\mathbf{ZWZ}^\top$. In Equation (14), the first and second terms are jointly denoted as *anchor regularization*, which encourage the learned $\tilde{\mathbf{S}}$ to maintain similarity with both the original $\mathbf{S}$ and the anchor topology. The third term, denoted as *sparsity regularization*, though appearing to minimize the nuclear norm of $\mathbf{W}$, essentially sparsifies $\tilde{\mathbf{S}}$, since $||\tilde{\mathbf{S}}||_* = ||\mathbf{W}||_*$ holds due to $\mathbf{Z}^\top \mathbf{Z} = \mathbb{I}_{r \times r}$. Therefore, Equation (14) achieves two key goals: (1) producing a sparse, refined communication topology, and (2) constraining the design to remain grounded in practical intuition. The resulting communication can be represented as follows:

$$\mathcal{G}_{com} = (\mathcal{V}, \mathcal{E}_{com}), \mathcal{E}_{com} = \{(i,j) \mid \tilde{\mathbf{S}}_{ij} \neq 0 \land (i,j) \in \mathcal{E}\}. \tag{15}$$

At this stage, we have successfully distilled a lightweight and informative collaboration network $\mathcal{G}_{com}$ from the sketched task-specific network $\tilde{\mathcal{G}}$, which is now ready to guide inter-agent message passing in the following process.

### 4.3. Optimizing G-Designer

Upon obtaining $\mathcal{G}_{com}$, the multi-agent utterances and dialogues can proceed as usual using $\mathcal{G}_{com}$, as detailed in Section 3.2. After $K$ rounds of interaction, the agents converge to a final solution $a^{(K)} = \mathcal{G}_{com}(\mathcal{Q})$. We then give the following optimization objective:

$$\underset{\Theta_e, \Theta_d}{\arg\min} \ \mathbb{E}_{\Theta_e, \Theta_d \sim \Omega} \Big[ u\big(\mathcal{G}_{com}(\mathcal{Q})\big) \Big], \tag{16}$$

where $\Theta_e$ and $\Theta_d$ are the parameters of the encoder $q(\cdot)$ and decoder $p(\cdot)$, respectively, $\Omega$ is the parameter space and $\mathbb{E}(\cdot)$ denotes the mathematical expectation. Equation (16) aims to maximize the utility of the generated solution, but it is inherently intractable and non-differentiable, as $u(\cdot)$ often depends on external API calls (Li et al., 2023b; Hendrycks et al., 2021). To address this, following standard approaches in multi-agent structure design (Zhuge et al., 2024; Zhang et al., 2024), we apply policy gradient (Williams, 1992) to approximate and optimize Equation (16):

$$\nabla_\Theta \mathbb{E}_{\Theta \sim \Omega} \Big[ u\big(\mathcal{G}_{com}(\mathcal{Q})\big) \Big] \approx \frac{1}{M} \sum_{k=1}^{M} u(a_m^{(K)}) \nabla_\Theta (P(\mathcal{G}_k)), \tag{17}$$

where $\mathbf{\Theta} = \{\Theta_e, \Theta_d\}$, $\{\mathcal{G}_k\}_{m=1}^{M}$ are indepently samples from $\mathcal{G}_{com}$, and $\{a_m^{(K)}\}_{m=1}^{M}$ are the corresponding output. $P(\mathcal{G}_k)$ calculates the probability of $\mathcal{G}_k$ being sampled, which can be expressed as $P(\mathcal{G}_k) = \prod_{i=1}^{N} \prod_{j=1}^{N} \tilde{\mathbf{S}}_{ij}$. Through iterative optimization guided by Equations (14) and (16) over a limited set of queries as the "training set", G-Designer efficiently develops task-awareness and the ca-

pability to strategically design the agent network, achieving truly task-customized multi-agent topology design.

**Optimization configuration**    The overall training objective of our method is formulated as $\mathcal{L}_{\text{G-Designer}} = \mathcal{L}_{utility} + \mathcal{L}_{anchor} + \mathcal{L}_{sparse}$, where $\mathcal{L}_{utility}$ represents the optimization target from Equation (16), $\mathcal{L}_{anchor}$ corresponds to the first and third terms in Equation (14), and $\mathcal{L}_{sparse}$ is the second term. Given a benchmark $\{\mathcal{Q}_i\}_{i=1}^{D}$ consisting of $B$ queries, G-Designer begins by optimizing with a small subset of $B'$ queries and fixes the learned parameters for testing on the remaining $(B - B')$ queries. The whole algorithm workflow of G-Designer is depicted in Algorithm 1.

## 5. Experiments

### 5.1. Experimental Setup

**Datasets and Metrics**    We evaluate G-Designer on three categories of datasets: ■ **General Reasoning**: MMLU (Hendrycks et al., 2021); ■ **Mathematical Reasoning:** GSM8K (Cobbe et al., 2021), MultiArith (Roy & Roth, 2016), SVAMP (Patel et al., 2021), and AQuA (Ling et al., 2017); ■ **Code:** HumanEval (Chen et al., 2021). We include the dataset statistics in Table 4.

**Baselines**    For single-agent approaches, we select **COT** (Wei et al., 2022), **ComplexCoT** (Fu et al., 2022), **Self-Consistency** (Wang et al., 2023a), and **PHP** (Zheng et al., 2023). For multi-agent topologies, we select **Chain**, **Star**, and **Tree** (formally defined in (Qian et al., 2024)), **Complete Graph** and **Random Graph** , **AutoGen** (Wu et al., 2023), **MetaGPT** (Hong et al., 2023), **LLM-Debate** (Du et al., 2023), **LLM-Blender** (Jiang et al., 2023), **DyLAN** (Liu et al., 2023), and **GPTSwarm** (Zhuge et al., 2024).

**Implementation Details**    We access the GPT via the OpenAI API, and mainly test on `gpt-4-1106-preview` (gpt-4) and `gpt-3.5-turbo-0125` (gpt-3.5). We set `temperature` to 0 for the single execution and single agent baselines and 1 for multi-agent methods. We set a summarizer agent to aggregate the dialogue history and produce the final solution $a^{(K)}$, with $K = 3$ across all experiments. The $\text{NodeEncoder}(\cdot)$ is implemented using `all-MiniLM-L6-v2` (Wang et al., 2020), with the embedding dimension set to $D = 384$. The anchor topology $\mathbf{A}_{anchor}$ is predefined as a simple chain structure. The sampling times $M$ are set as 10, and $\tau = 1e-2$ and $\zeta = 1e-1$ are set for all experiments. We provide explicit agent profiling for multi-agent methods, following the classical configurations in LLM-MA systems (Liu et al., 2023; Zhuge et al., 2024; Yin et al., 2023), and use `gpt-4` to generate agent profile pools. For all benchmarks, we merely use $B' \in \{40, 80\}$ queries for optimization.

### 5.2. Main Results

In this section, we conduct extensive experiments across six benchmarks to verify that G-Designer is:

*Table 1.* Performance comparison with three types of baselines, including single-agent execution, spatial communication, and temporal communication. The best results are in bold, and the runner-ups are underlined. All methods, except for the single-agent category, utilize **five** `gpt-4`-based agents. "Mul.", "Ada.", and "Rob." indicate whether the method supports a multi-agent setting, whether it is task-adaptive, and whether it is adversarially robust, respectively. ✗, ✔ and ✓ signifies no/partial/full support in these aspects.

| Method | Mul. | Ada. | Rob. | MMLU | GSM8K | MultiArith | SVAMP | AQuA | HumanEval | Avg. |
|---|---|---|---|---|---|---|---|---|---|---|
| Vanilla | ✗ | ✗ | ✗ | 82.14 | 85.40 | 93.15 | 87.18 | 70.34 | 71.68 | 81.65 |
| CoT | ✗ | ✗ | ✗ | 82.65↑0.51 | 87.17↑1.77 | 94.79↑1.64 | 88.32↑1.14 | 73.91↑3.57 | 75.52↑3.84 | 83.73 |
| ComplexCoT | ✗ | ✗ | ✗ | 83.78↑1.64 | 87.62↑2.22 | 95.86↑2.71 | 90.17↑2.99 | 77.58↑7.24 | 74.94↑3.26 | 84.99 |
| SC (CoT) | ✗ | ✗ | ✗ | 82.66↑0.52 | 87.93↑2.53 | 96.88↑3.73 | 88.69↑1.51 | 75.08↑4.74 | 77.30↑5.62 | 84.75 |
| SC (ComplexCoT) | ✗ | ✗ | ✗ | 83.65↑1.51 | 86.14↓0.74 | 96.94↑3.79 | 89.72↑2.54 | 77.69↑7.35 | 77.94↑6.26 | 85.35 |
| PHP | ✓ | ✗ | ✗ | 83.45↑1.31 | **95.50**↑10.1 | 98.10↑2.84 | 90.02↑3.44 | 79.00↑8.66 | 82.96↑11.36 | 88.17 |
| Chain | ✓ | ✗ | ✗ | 82.35↑0.21 | 85.57↑0.17 | 94.38↑1.23 | 83.41↓3.77 | 70.94↑0.60 | 80.88↑9.20 | 82.92 |
| Star | ✓ | ✗ | ✗ | 80.79↓1.35 | 85.55↑0.15 | 93.79↓0.64 | 88.09↓0.91 | 68.57↓1.77 | 75.65↑3.97 | 82.07 |
| Tree | ✓ | ✗ | ✗ | 81.89↓0.25 | 84.56↓0.84 | 94.60↑1.45 | 89.25↑2.07 | 72.84↑2.50 | 77.38↑5.70 | 83.42 |
| Complete Graph | ✓ | ✗ | ✗ | 83.15↑1.01 | 86.49↑1.09 | 97.20↑4.05 | 89.48↑2.30 | 79.21↑8.87 | 83.75↑12.07 | 86.55 |
| Random Graph | ✓ | ✗ | ✗ | 83.76↑1.62 | 86.14↑0.74 | 95.46↑2.31 | 85.41↓1.77 | 74.07↑3.73 | 82.66↑10.98 | 84.58 |
| AutoGen | ✓ | ✗ | ✗ | 82.13↓0.01 | 90.06↑7.92 | 93.80↑0.65 | 88.44↓1.26 | 73.65↑3.31 | 85.41↑13.73 | 85.58 |
| MetaGPT | ✓ | ✗ | ✗ | - | - | - | - | - | 85.90↑14.22 | 84.90 |
| LLM-Blender | ✓ | ✗ | ✗ | 81.22↓0.92 | 89.17↑3.77 | 94.27↑1.12 | 88.77↑1.59 | 77.05↑6.71 | - | 86.09 |
| LLM-Debate | ✓ | ✗ | ✓ | 83.69↑1.55 | 90.23↑4.83 | 96.27↑3.12 | 90.56↑3.38 | 77.52↑7.18 | 83.79↑12.11 | 87.01 |
| DyLAN | ✓ | ✔ | ✓ | 80.16↓1.98 | 88.16↑2.76 | 94.27↑1.12 | 87.40↑0.22 | 74.16↑3.82 | 89.70↑18.02 | 85.64 |
| GPTSwarm | ✓ | ✔ | ✓ | 83.98↑1.84 | 89.74↑4.34 | 97.84↑4.69 | 86.42↓0.76 | 78.16↑7.82 | 88.49↑16.81 | 87.32 |
| G-Designer | ✓ | ✓ | ✓ | **84.50**↑2.36 | 95.07↑9.67 | **98.30**↑5.15 | **91.85**↑4.67 | **79.47**↑9.13 | **89.90**↑18.22 | **89.84** |

**High-performing** The experimental results from Table 1 demonstrate that `G-Designer` is effective in designing powerful LLM-MA topologies. Concretely, `G-Designer` achieves the best performance in five out of six benchmarks, and on GSM8K, it trails only PHP with a 9.67% ↑ accuracy improvement. On the HumanEval benchmark, `G-Designer` surpasses MetaGPT, a specialized multi-agent code generation framework, by 4.0% at *pass@1*, and outperforms state-of-the-art multi-agent collaboration frameworks like GPTSwarm and DyLAN by margins of 0.20% ∼ 1.41%.

**Task-adaptive** Figure 6 visualizes the different topologies designed by `G-Designer` for varying query difficulties on HumanEval and GSM8K. As shown in Figure 6, the multi-agent topologies generated by `G-Designer` are highly dependent on the specific task context and its difficulty. In *Case a*, despite having five `gpt-4` agents available as design resources, `G-Designer` identified the task of designing a `strlen(string)` function as relatively simple. It streamlined the topology by removing unnecessary agents and retained only a minimal "Algorithm Designer → Programmer" structure to solve the problem. In contrast, for the more complex *Case c* and *Case e*, `G-Designer` crafted a more intricate communication graph. These cases highlight the strong task-adaptiveness of `G-Designer`.

**Scalable** To evaluate the scalability of `G-Designer` to a larger number of agents, we report its performance across 5 ∼ 20 agents, as presented in Table 6. Notably, `G-Designer` exhibits a steeper performance gain than GPTSwarm as the agent count increases. More importantly,

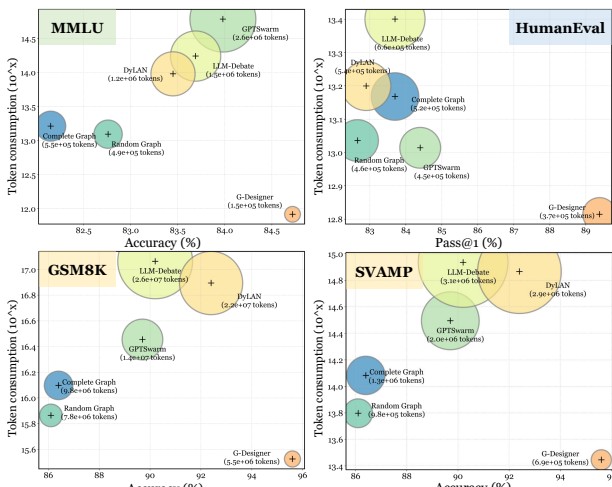

*Figure 4.* Visualization of the performance metrics and prompt token consumption of different multi-agent communication topologies across MMLU, HumanEval, GSM8K, and SVAMP. The diameter of each point is proportional to its $y$-axis value.

while the complete graph and GPTSwarm incur an overwhelming token cost at 20 agents (5.6 ∼ 30.3M tokens), `G-Designer` achieves superior performance with merely 6.11% of GPTSwarm's prompt token consumption, surpassing it by 2.44% ↑. These results decisively demonstrate the scalability and potential of `G-Designer` in advancing large-scale autonomous multi-agent systems.

**Token-economical (*inference*)** A key benefit of `G-Designer`'s adaptivity is that it prevents the use of overly complex structures for simple tasks, thus minimizing

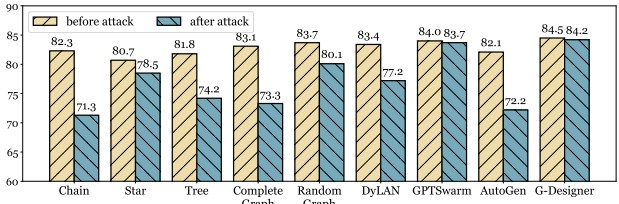

*Figure 5.* We compare the accuracy (%) of various multi-agent frameworks before and after prompt attacks on MMLU.

unnecessary communication costs—in the case of LLM-MA, reducing token consumption. Figure 4 It illustrates the differences in prompt token consumption between `G-Designer` and several representative multi-agent designs. We observe that simpler topologies, such as complete graphs and random graphs, consume fewer tokens but show significantly weaker performance. More complex communication structures, like GPTSwarm and DyLAN, achieve superior performance, albeit at the cost of excessive token consumption. For instance, DyLAN's cost on GSM8K is $2.82\times$ that of the random graph, reaching a substantial $2.2e + 7$. In contrast, `G-Designer` elegantly balances both efficiency and task performance, achieving the highest performance across all four benchmarks while maintaining the lowest token cost. For example, on SVAMP, `G-Designer` surpasses DyLAN by $4\%$ while using only $23.7\%$ of DyLAN's token cost.

**Resource-efficient (*training*)** We validate `G-Designer`'s **training process is resource-friendly** from three dimensions: GPU cost, token cost, and wall-clock time. Table 5 showcases that training `G-Designer` with up to 1000 agents requires less than 4GB of memory. Table 2 unveils that `G-Designer` not only attains the highest accuracy but also exhibits superior token efficiency and reduced wall-clock time compared to existing baselines, underscoring its effectiveness in multi-agent collaboration.

*Table 2.* Efficiency analysis. We compare the training/inference wall-clock time and token consumption between `G-Designer` and other high-performing baselines on the GSM8K dataset.

| Method | Perf. | #Training Token | #Inference Token | #Overall Token | Training Time | Inference Time |
|---|---|---|---|---|---|---|
| Complete | 86.4 | - | $9.8 \times 10^6$ | $9.8 \times 10^6$ | - | 2.4h |
| DyLAN | 88.1 | $9.6 \times 10^6$ | $1.3 \times 10^7$ | $2.2 \times 10^7$ | 2.8h | 4.6h |
| GPTSwarm | 89.7 | $5.5 \times 10^6$ | $8.4 \times 10^6$ | $1.4 \times 10^7$ | 2.1h | 2.8h |
| `G-Designer` | **95.0** | $\mathbf{2.7 \times 10^5}$ | $\mathbf{8.2 \times 10^6}$ | $\mathbf{8.5 \times 10^6}$ | **0.3h** | **2.3h** |

### 5.3. Robustness Analysis

Following (Zhuge et al., 2024), we simulate a system prompt attack on one of the five agents. As seen in Figure 5, many trivial structures, such as chain or complete graph, experience significant performance degradation under partial system attacks, with drops as high as $11.0\%$. Among more sophisticated structures, GPTSwarm, benefiting from its specialized node optimization mechanism, only suffers a minor $0.3\%$ accuracy decline. However, other methods fare less

well, with DyLAN and AutoGen showing accuracy drops of $6.2\%$ and $9.9\%$, respectively. Remarkably, `G-Designer` demonstrates exceptional robustness against adversarial attacks, maintaining nearly identical performance pre- and post-attack. This resilience can be attributed to its agent encoding capability, which, during optimization, can detect malicious inputs and prune the corresponding edges.

### 5.4. Framework Analysis

**Ablation Study.** We report results for two variants of ourmethod: **(1) *w/o* SR**, which removes the sparsity regularization in Equation (14), **(2) *w/o* Anchor**, which excludes the anchor structure $\mathbf{A}_{anchor}$, **(3) *w/o* NodeEncoder**, removing node encoder in Equation (7), and **(4) *w/o* $v_{\text{task}}$** in Equation (8). As shown in Table 3, removing the task virtual node disrupts `G-Designer`'s task-adaptiveness, leading to the most significant performance drop. The removal of $\mathbf{A}_{anchor}$ consistently leads to performance degradation, while the absence of sparsity regularization makes the system more vulnerable to adversarial attacks.

**Discussion on anchor topology.** Given that `G-Designer` is initialized with the anchor topology $\mathbf{A}_{\text{anchor}}$ introduced in Section 4.1, one may question whether the performance gains of `G-Designer` primarily stem from $\mathbf{A}_{\text{anchor}}$ itself. In response, we emphasize that the anchor topology corresponds to the simple Chain structure in Table 1, where `G-Designer` achieves substantial improvements over it, specifically $9.50\% \uparrow$ on GSM8K and $8.44\% \uparrow$ on SVAMP. Thus, we assert that the superior performance of `G-Designer` is predominantly attributed to its adaptive topology design rather than the anchor topology itself.

## 6. Conclusion

In this paper, we first present the LLM-based Multi-agent Communication Protocol (MACP), which aims to provide insightful guidance for designing complex multi-agent systems. Furthermore, we propose an effective, adaptive, and robust LLM-powered multi-agent communication graph designer, termed `G-Designer`, to facilitate the automated design of collaborative AI systems. `G-Designer` is highly task-aware, dynamically crafting compact and robust communication topologies based on the complexity of the task at hand. We hope that `G-Designer` will inspire future research

*Table 3.* Ablation study of `G-Designer`'s four variants, tested on MMLU benchmark.

| Variant | MMLU | | GSM8K | |
|---|---|---|---|---|
| | Clean | Attack | Clean | Attack |
| vanilla `G-Designer` | 84.5 | 84.2 | 95.0 | 92.5 |
| *w/o* SR | 84.1 | 83.2 | 94.4 | 90.7 |
| *w/o* Anchor | 84.0 | 83.8 | 94.7 | 92.0 |
| *w/o* NodeEncoder$(\cdot)$ | 83.2 | 82.4 | 92.8 | 87.4 |
| *w/o* $v_{\text{task}}$ | 81.3 | 82.0 | 90.3 | 87.7 |

on the emergence of self-organizing and self-evolving collective intelligence.

## Acknowledgements

The work is supported by the National Science Foundation of China (62472317).

## Impact Statement

**Ethical impacts.** We confidently affirm that our paper is free of ethical concerns across its motivation, design, experiments, and data usage. The proposed `G-Designer` method aims to advance the fields of multi-agent systems and communication topologies design automation, contributing positively and responsibly to the scientific community.

**Expected societal implications.** Our paper presents a significant advancement in multi-agent systems. `G-Designer` addresses the challenge of choosing the right communication topology for specific tasks. `G-Designer` not only provides a practical solution for multi-agent deployment but also paves the way for future studies on collective intelligence systems. Furthermore, as long as the base LLM used is aligned with human values, our system will not generate harmful content.

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

# A. Algorithm Workflow

---

**Algorithm 1** Designing workflow of `G-Designer`

---

**Input** : Input query $\mathcal{Q}$, Graph auto-encoder $f_v$ composed of encoder $q(\cdot)$ and decoder $p(\cdot)$ (parameterized by $\Theta_e$ and $\Theta$), learning rate $\alpha$

**for** *query d in* $\{1, 2, \cdots, D'\}$ **do**

   /* Establish multi-agent network                                                    */

   **for** *node i in* $\{1, 2, \cdots, N\}$ **do**

      $\mathbf{x}_i \leftarrow \text{NodeEncoder}\left(\mathcal{T}(\texttt{Base}_i), \texttt{Role}_i, \mathcal{T}(\texttt{Plugin}_i)\right)$

   **end**

   Obtain agent embeddings $\mathbf{X}_{agent} \leftarrow [\mathbf{x}_1, \mathbf{x}_2, \cdots, \mathbf{x}_N]^\top$

   Obtain task-specific node $x_{task} \leftarrow \text{NodeEncoder}(\mathcal{Q}_d)$

   Set an anchor topology $\mathbf{A}_{anchor}$ // In our experiments, the anchor topology is simply set as the chain structure

   Obtain a task-specific multi-agent network $\tilde{\mathcal{G}} = \left(\begin{bmatrix} \mathbf{X}_{agent} \\ \mathbf{x}_{task}^\top \end{bmatrix}, \mathbf{A}_{anchor}\right)$ // Note that $\mathbf{A}_{anchor}$ here contains bidirectional edges added by the task node $v_{task}$

   /* Design communication topology                                           */

   Encode $\tilde{\mathcal{G}}$ into latent agent representations $\mathbf{H}_{agent}$: $q(\mathbf{H}_{agent} \mid \tilde{\mathbf{X}}, \mathbf{A}_{anchor}) = \prod_{i=1}^{N} q(\mathbf{h}_i \mid \tilde{\mathbf{X}}, \mathbf{A}_{anchor})$

   Decode (phase 1) and obtain the sketched graph $\mathbf{S}$: $p_s(\mathbf{S} \mid \mathbf{H}_{agent}) = \prod_{i=1}^{N} \prod_{j=1}^{N} p_s(\mathbf{S}_{ij} \mid \mathbf{h}_i, \mathbf{h}_j, \mathbf{h}_{task}; \Theta_d)$,

   Decode (phase 2) and obtain the communication graph $\mathcal{G}_{com} = (\mathcal{V}, \mathcal{E}_{com}), \mathcal{E}_{com} = \{(i,j) \mid \tilde{\mathbf{S}}_{ij} \neq 0 \wedge (i,j) \in \mathcal{E}\})$

   /* Guide multi-agent system collaboration                                */

   **for** *iteration t in* $\{1, 2, \cdots, K\}$ **do**

      **for** *node i in* $\phi(\mathcal{G}_{com})$ **do**

         Agent $v_i$ generates $\mathcal{R}_i^{(t)} \leftarrow v_i(\mathcal{P}_{\text{sys}}^{(t)}, \mathcal{P}_{\text{usr}}^{(t)}), \; \mathcal{P}_{\text{usr}}^{(t)} = \{\mathcal{Q}, \cup_{v_j \in \mathcal{N}_{\text{in}}(v_i)} \mathcal{R}_j^{(t)}\}$

      **end**

      /* Aggregate solution                                                            */

      $a^{(t)} \leftarrow \text{Aggregate}(\mathcal{R}_1^{(t)}, \mathcal{R}_2^{(t)}, \cdots, \mathcal{R}_N^{(t)})$

   **end**

   /* Update `G-Designer` parameters                                               */

   $\Theta^{d+1} \leftarrow \Theta^d - \alpha \nabla_{\Theta^d} \mathcal{L}_{\texttt{G-Designer}}$

**end**

---

# B. Dataset Statistics

We conclude the dataset statistics in Table 4.

# C. Case Study

Figure 6 visualizes the different topologies designed by `G-Designer` for varying query difficulties on the HumanEval and GSM8K benchmarks.

# D. Supplementary Results

*Table 4.* Dataset descriptions and statistics.

| Category | Dataset | Answer Type | Metric | #Test | License |
|---|---|---|---|---|---|
| General reasoning | MMLU | Multi-choice | Acc. | 153 | MIT License |
| Math reasoning | GSM8K | Number | Acc. | 1,319 | MIT License |
| | MultiArith | Number | Acc. | 600 | Unspecified |
| | SVAMP | Number | Acc. | 1,000 | MIT License |
| | AQuA | Multi-choice | Acc. | 254 | Apache-2.0 |
| Code generation | HumanEval | Code | Pass@1 | 164 | MIT License |

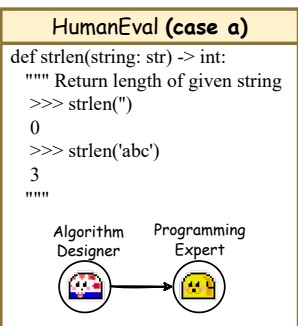

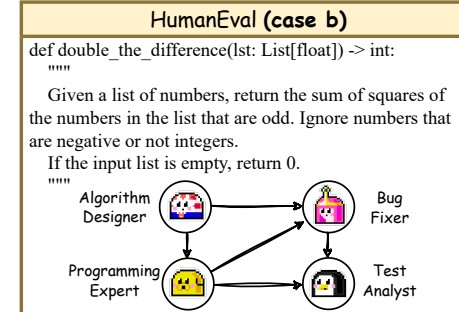

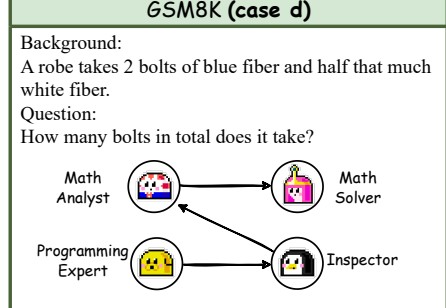

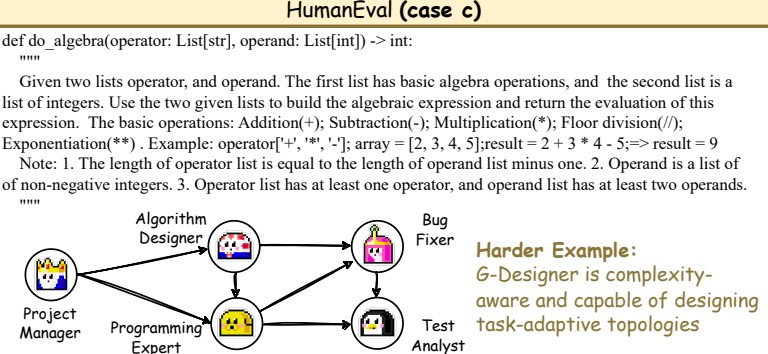

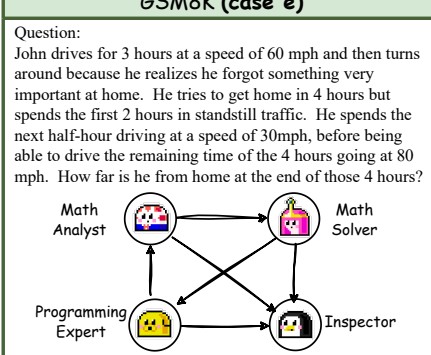

*Figure 6.* Case study of the communication topologies designed by `G-Designer` on HumanEval and GSM8K benchmarks.

*Table 5.* The GPU cost of `G-Designer` with increasing number of agents.

| #Agents | 5 | 50 | 100 | 1000 |
|---|---|---|---|---|
| Memory (GB) | 2.7 | 2.9 | 3.0 | 3.8 |

*Table 6.* Comparison of accuracy, time, token consumption, and cost across different agent configurations. We use the MMLU benchmark and `gpt-3.5-turbo` as the base LLM.

| #Agents | 5 | 10 | 20 |
|---|---|---|---|
| **Chain** | | | |
| Accuracy (%) | 70.59 | 71.24 | 71.98 |
| Time (min) | 15.73 | 30.20 | 56.18 |
| #Prompt Tokens | 351,802 | 702,164 | 1,378,328 |
| Cost (USD) | 0.5228 | 1.0434 | 2.0482 |
| **Complete Graph** | | | |
| Accuracy (%) | 71.90 | 72.16 | 72.51 |
| Time (min) | 16.85 | 34.21 | 66.47 |
| #Prompt Tokens | 545,984 | 1,669,451 | 5,648,834 |
| Cost (USD) | 0.7161 | 2.1770 | 7.3662 |
| **GPTSwarm** | | | |
| Accuracy (%) | 72.55 | 73.86 | 75.38 |
| Time (min) | 62.14 | 186.86 | 412.18 |
| #Prompt Tokens | 3,055,236 | 9,048,465 | 30,317,341 |
| Cost (USD) | 4.2190 | 12.4961 | 41.4235 |
| `G-Designer` | | | |
| Accuracy (%) | **73.20** | **74.51** | **77.82** |
| Time (min) | 19.26 | 36.04 | 68.89 |
| #Prompt Tokens | 452,329 | 885,332 | 1,852,538 |
| Cost (USD) | 0.6036 | 1.2768 | 2.6713 |

