# OpenReview forum: "G-Designer: Architecting Multi-agent Communication Topologies via Graph Neural Networks"
_ICML.cc/2025/Conference — ICML 2025 spotlightposter_

### Official Review · Reviewer_e1AK · 2025-03-05

**Overall Recommendation:** 4

**Summary:**

Recent research has shown that large language model (LLM)-based multi-agent systems outperform traditional single-agent methods. This paper focuses on the challenge of choosing effective communication topologies for multi-agent systems. It puts forward the LLM-based Multi-agent Communication Protocol (MACP), which serves as a guiding principle for topology design, and presents G-Designer, an adaptable solution for multi-agent deployment.
G-Designer represents the multi-agent system as a graph. It uses a variational graph auto-encoder to encode agents, along with a task-specific virtual node, and then decodes a communication topology that is tailored to the task at hand. The study finds that the performance of different topologies varies with task complexity.
Experimental evaluations across six benchmarks verify that G-Designer exhibits high performance, task adaptability, and adversarial robustness. In conclusion, G-Designer offers a practical and efficient approach for multi-agent system deployment, paving the way for further research in collective intelligence systems. ## update after rebuttal

**Claims And Evidence:**

This paper mainly puts forward the following ideas. All of these claims seem to be adequately supported:

1. It posits that LLM-based multi-agent systems can surpass traditional single-agent approaches. Moreover, the topologies of multi-agent systems can be optimized to yield superior outcomes across a variety of tasks. This assertion is substantiated by prior research and the experiments conducted in this study.
2. The paper regulates multi-agent topology design across three dimensions: effectiveness, complexity-adaptiveness, and adversarial robustness. These logics are validated through in-depth analysis, demonstrating their effectiveness in optimizing communication structures.
3. The study introduces G-Designer, a novel solution for multi-agent deployment. G-Designer models the multi-agent system as a graph and employs a variational graph auto-encoder to encode agents and a task-specific virtual node. The algorithmic details of G-Designer are clearly presented and supported by theoretical derivations and experimental results.

**Essential References Not Discussed:**

To the best of my knowledge, all relevant related works that should be mentioned have been included.

**Experimental Designs Or Analyses:**

The experimental design of the paper is thorough and comprehensive. The authors conducted extensive experimental evaluations of G-Designer across six benchmarks, validating its high performance, task adaptability, and adversarial robustness in multi-agent systems. It is worth noting that the authors performed adversarial attack experiments to verify the robustness of G-Designer, which expands the application prospects of the method. This ensures that in scenarios where there are poisoned agents or less intelligent agents, G-Designer can detect them and maintain usability.

**Methods And Evaluation Criteria:**

I believe that the methods and ideas proposed in this paper hold significant practical value for the advancement and application of multi-agent systems. The article introduces a novel multi-agent communication protocol based on large language models, termed the LLM-based Multi-agent Communication Protocol (MACP), which can serve as a guiding principle for the topological design of multi-agent systems. Furthermore, the paper presents an effective, adaptive, and robust LLM-driven multi-agent communication graph designer, G-Designer, to facilitate the automated design of collaborative AI systems. For me, G-Designer has the potential to inspire future research on the emergence of self-organizing and self-evolving collective intelligence.

**Other Comments Or Suggestions:**

Typos:
1. In Table 1, the average accuracy of `MetaGPT` appears to be miscalculated.
2. In Figure 3, 'Ouput solution' -> 'Output solution'.

**Other Strengths And Weaknesses:**

1. The methodology is innovative. While the abstraction of multi-agent collaboration into graph structures has been mentioned in previous papers, this paper introduces VGAE on top of that. The combination of these two aligns with intuition and leverages the powerful graph representation capabilities of GNNs, making it highly novel.
2. The paper is clearly articulated. Particularly, the illustrations and case studies are visually appealing.
3. The training burden is relatively light. I observed that the experimental section not only discusses the effectiveness and efficiency during inference but also mentions the training overhead in `Table 2`. This is crucial for large-scale applications, as excessive training costs can limit the model's applicability. The overhead of this model appears to be lower compared to previous methods.
However:
1. The paper does not seem to provide specific examples of agent profiles. I am aware that the profile generation method in the paper follows the classical configurations in LLM-MA systems. However, I am curious about the impact of different qualities and contents of profiles on the results. Is the dynamic collaboration graph design sufficiently adaptable to different profiles?

**Questions For Authors:**

See weaknesses.

**Relation To Broader Scientific Literature:**

I believe this paper can be primarily linked to the Agent Scaling Law mentioned in Macnet[1]. Macnet suggests that as the number of agents increases, the performance of multi-agent systems gradually improves, but this improvement is associated with the collaboration patterns among the agents. The G-Designer proposed in this paper can automatically design appropriate collaboration patterns based on the complexity of the task, thereby better adapting to the requirements of different tasks and enhancing the performance of multi-agent systems. Therefore, I think applying G-Designer to a larger number of agents could potentially better validate the effectiveness of the Agent Scaling Law.
[1] Scaling Large Language Model-based Multi-Agent Collaboration ICLR'25

**Theoretical Claims:**

I have reviewed the formulas and derivations in Section 4 and found no explicit flaws. Formula (9) employs the variational graph auto-encoder (VGAE) framework to generate the multi-agent interaction topology. Formula (10) presents the specific form of the encoder module within the VGAE, where the innovative introduction of a task-specific virtual node and agent nodes, along with the use of GNN to learn the representations of agent nodes, results in a reasonable representation of the agent collaboration graph.

---

> ### Author Rebuttal · Authors · 2025-03-31
>
> We sincerely thank you for your careful comments and thorough understanding of our paper! Here we give point-by-point responses to your comments and describe the revisions we made to address them.
>
> ---
> > **`Weakness 1`**: Would switching between different agent roles have an impact on the experimental results?
>
> Thank you for the insightful comment! To address your concerns, we analyze the transferability of G-Designer to unseen agent roles as follows:
>
> *Table A. Generalization of G-Designer to unseen agent roles. We train G-Designer on the "dataset for optimization" and evaluate it on the "dataset for test," where the test set may include agent roles unseen during training.*
>
> | Dataset for Optimization | Roles | Dataset for Test | (Unseen) Roles | Performance |
> |--------------------------|-------|------------------|----------------|-------------|
> | GSM8K | - | GSM8K | - | 95.07 |
> | HumanEval | Project Manager, Algorithm Designer, etc. | GSM8K | Math Solver, Mathematical Analyst, etc. | 93.98 |
> | HumanEval | - | HumanEval | - | 89.90 |
> | MMLU | Knowledgeable Expert, Mathematician, etc. | HumanEval | Project Manager, Algorithm Designer, etc. | 88.72 |
>
> The results indicate that G-Designer demonstrates strong transferability across datasets, effectively adapting from HumanEval to GSM8K and from MMLU to HumanEval, among others.
>
> ---
> > **`Weakness 2`**: Typos
>
> Thank you very much for your careful review of our manuscript. We have carefully addressed the typos you pointed out and made the necessary corrections throughout the document. Your feedback has been very helpful in improving the clarity and accuracy of the paper.
>
> We appreciate your attention to detail and your valuable contribution to the review process.

---

> > ### Comment · Reviewer_e1AK · 2025-04-04
> >
> > I have read the author's response and I will maintain my score

---

> > > ### Author Response · Authors · 2025-04-07
> > >
> > > **Dear Reviewer e1AK**,
> > >
> > > Thank you for your detailed, thoughtful feedback and for taking the time to review our responses. We truly appreciate your constructive insights and your willingness to engage with our clarifications. Your recognition of the **innovative implementation of G-Designer** and of its **lightweight, resource-saving design**, has truly inspired us!
> > >
> > > Best Regards,
> > >
> > > Authors

---

### Official Review · Reviewer_gFC8 · 2025-03-09

**Overall Recommendation:** 4

**Summary:**

This paper introduces G-Designer, an framework for designing task-aware, adaptive, and robust communication topologies for multi-agent systems powered by large language models (LLMs). G-Designer dynamically generates communication structures for specific user requests using a variational graph auto-encoder (VGAE), which encodes both agent profiles and task-specific information to decode an optimal interaction graph. This method aims to balance performance, adaptability, and robustness by minimizing communication overhead while maintaining high accuracy and resilience against adversarial attacks.

**Claims And Evidence:**

The main claims by this paper (i.e., task-awareness, adaptiveness, robustness, high-performance) are well supported by the methodology and experiments. I particularly appreciate the emphasis on task-adaptiveness in G-Designer, a feature that has been largely absent in prior LLM-based multi-agent systems. Systems such as MetaGPT and ChatDev adhere to static SOPs, and MacNet exhibits poor scaling capabilities due to its fixed structure. Although GPTSwarm incorporates dynamic training, it relies on a fixed topology during testing. In the latest ICLR 2025 papers, works like AFlow, AgentPrune, and ADAS fail to dynamically allocate inference resources based on task complexity. Therefore, I think G-Designer’s advocacy for task-adaptiveness and its proposed Multi-agent Communication Protocol (MACP) represent a significant contribution that warrants attention from the research community.

**Essential References Not Discussed:**

I recommend that the authors include the following recent multi-agent papers. Incorporating these references would further emphasize the novelty of their contribution:

[1] AFlow: Automating Agentic Workflow Generation, ICLR 2025

[2] AgentSquare: Automatic LLM Agent Search in Modular Design Space, ICLR 2025

[3] Automated Design of Agentic Systems, ICLR 2025

[4] Flow: Modularized Agentic Workflow Automation, ICLR 2025

[5] Talk Structurally, Act Hierarchically: A Collaborative Framework for LLM Multi-Agent Systems

[6] Model Swarms: Collaborative Search to Adapt LLM Experts via Swarm Intelligence

**Experimental Designs Or Analyses:**

G-Designer is compared against vanilla LLM, single-agent, and multi-agent methods. In the cost-efficiency scatter plot (Figure 4), G-Designer achieves the best performance with the lowest token cost among all methods. Additionally, I highly appreciate the case study and scalability study presented in Figure 6 and Table 6, which I strongly suggest be moved to the main text. Figure 6 provides an intuitive understanding of G-Designer’s operational logic, and Table 6 suggests that G-Designer may offer preliminary insights into agent scaling laws. Current graph-based multi-agent systems, such as MacNet and GPTSwarm, suffer from poor scaling laws, with marginal performance improvements as the number of agents increases (MacNet is especially criticized by reviewers for this issue). G-Designer explicitly highlights this challenge, namely the quadratic growth in communication edges, for which the authors propose a lightweight solution.

**Methods And Evaluation Criteria:**

The authors primarily employed three categories of benchmarks. Overall, the results on these benchmarks support the superiority of their approach. However, I recommend that the authors include more agent-specific benchmarks, such as API-Bank [1] and AgentBench [2].

[1] API-Bank: A Comprehensive Benchmark for Tool-Augmented LLMs

[2] AgentBench: Evaluating LLMs as Agents

**Other Comments Or Suggestions:**

No specific comments.

**Other Strengths And Weaknesses:**

Strength:

 - The motivation and methodology presented in this paper are clear and accessible. Figure 3 is particularly well-designed and illustrative.

 - The proposed method is well-structured and methodologically rigorous, ensuring robustness, adaptability, and high performance.

Weakness:

 - The experiments were conducted only on GPT-4 and GPT-3.5, which are no longer state-of-the-art models. Are the authors planning to test more advanced or open-source LLMs?

 - The system prompt attack described in Section 5.3 requires more detailed elaboration. Moreover, there are more advanced agent attack methods available, such as those in [1]. Do the authors plan to evaluate G-Designer’s defense capabilities against a broader range of attacks?

[1] Agent Security Bench (ASB): Formalizing and Benchmarking Attacks and Defenses in LLM-based Agents

**Questions For Authors:**

- Do the authors plan to test more advanced or open-source LLMs?

 - In Section 5.3, do the authors intend to incorporate more advanced agent attack methods?

 - Are the authors considering using more agent-specific benchmarks to evaluate G-Designer?

 - G-Designer employs a chain-based anchor graph as its starting point. Do the authors plan to explore alternative anchor graph initialization?

**Relation To Broader Scientific Literature:**

This paper builds on the research trajectory of GPTSwarm, MacNet, and AgentPrune, which use DAGs to model multi-agent systems, and is highly relevant to the current line of research on (automated) LLM-based multi-agent systems.

**Theoretical Claims:**

No significant theoretical issues identified.

---

> ### Author Rebuttal · Authors · 2025-03-31
>
> We would like to express our deepest respect for your meticulous review! In response to your efforts, we have carefully prepared a point-by-point reply:
>
> ---
> > **`Weakness 1`**: Supplement more agent-specific benchmarks.
>
> Thank you very much for your insights. We conducted experiments on the more agent-specific GAIA benchmark to assess the effectiveness of our method. The experimental results are as follows:
>
> *Table A. Performance comparison on GAIA benchmark.*
> |Method|Level 1|Level 2|Level 3|Avg.|
> |-|-|-|-|-|
> |Vanilla GPT-4|9.68|1.89|2.08|4.05|
> |DyLAN|13.98|4.40|0|6.69|
> |GPTSwarm|23.66|16.35|2.04|16.33
> |**G-Designer**|**25.16**|**18.87**|**2.04**|**16.94**|
>
>
> This demonstrates that our method effectively outperforms baselines even in more agentic benchmarks like GAIA. We sincerely hope this addresses your concerns.
>
> ---
> > **`Weakness 2`**: Include the recent multi-agent papers.
>
> Your erudition and wisdom have been immensely helpful to us! We commit to including the papers you mentioned in the revised manuscript.
>
> ---
> > **`Weakness 3`**: Are the authors planning to test more advanced or open-source LLMs?
>
> To address your concern, we use the newer and popular closed-source model GPT-4o-mini and the open-source model DeepSeek-V3 as base models, and compare the experimental results of our method with Vanilla and GPTSwarm in Table B.
>
> *Table B. Performance results with more advanced LLMs.*
> | Method | LLM | MMLU | GSM8K | HumanEval |
> |:-:|:-:|:-:|:-:|:-:|
> | Vanilla | GPT-4o-mini | 77.12 | 92.67 | 85.71 |
> | Vanilla | DeepSeek-V3 | 86.27 | 93.25 | 87.67 |
> | GPTSwarm | GPT-4o-mini | 78.43 | 93.78 | 86.28 |
> | GPTSwarm | DeepSeek-V3 | 86.93 | 94.38 | 89.72 |
> | G-Designer | GPT-4o-mini | 79.73 | 94.23 | 90.32 |
> | G-Designer | DeepSeek-V3 | 89.54 | 95.52 | 91.93 |
>
> Due to the time constraints of the rebuttal, we were unable to compare our method with more baselines. However, the experimental results presented above demonstrate that our method is also applicable to newer LLMs.
>
> ---
> > **`Weakness 4`**: Do the authors plan to evaluate G-Designer’s defense capabilities against a broader range of attacks?
>
> To address your concerns, we evaluated G-Designer under the PoisonRAG [1] attack, with results presented in the table below:
>
> *Table C. Performance comparison on MMLU. We employ the configuration from PoisonRAG, inserting erroneous messages into the contextual memory of the attacker agent, enabling them to disseminate conclusions derived from these messages to others.*
> |Method|Chain|Star|Complete Graph|DyLAN|GPTSWarm|G-Designer|
> |-|-|-|-|-|-|-|
> |Before attack|82.3|80.7|83.1|83.4|84.0|84.5|
> |After attack|73.6|65.8|75.2|77.9|81.0|83.6|
>
> The results indicate that even under memory attacks, G-Designer effectively mitigates interference from malicious agents through its dedicated sparsification mechanism.
>
> [1] Poison-rag: Adversarial data poisoning attacks on retrieval-augmented generation in recommender systems.
>
>
> ---
> > **`Weakness 5`**: Do the authors plan to explore alternative anchor graph initialization?
>
>
> The suggestion you proposed is very thoughtful! In `Table C`, we explored the impact of different initial topologies on the experimental results.
>
> *Table D: Different Topologies for anchor graph initialization on GSM8K dataset.*
>
> |Anchor|Accuracy|
> |-|-|
> |Chain|95.07|
> |Layered|94.92|
> |Star|94.47|
> |Random|91.15|
> |FullConnected|95.22|
>
> Using different topologies as initial anchors shows minimal performance differences, although the cost is higher. This aligns with our analysis in Section 5.4, where we concluded that the success of G-Designer is mainly attributed to its efficient communication graph generation.

---

> > ### Comment · Reviewer_gFC8 · 2025-04-04
> >
> > I have read the author's rebuttal and the review of other reviewer. I'd love to maintain my accept score.

---

> > > ### Author Response · Authors · 2025-04-07
> > >
> > > **Dear Reviewer gFC8**,
> > >
> > > Thank you for your thoughtful feedback and continuous support of our work. Your comments helped us refine the presentation, enhance the experimentation, and strengthen the manuscript. We particularly appreciate your recognition of **the novelty of G-Designer** and **the importance of our proposed Multi-agent Communication Protocol (MACP) to the MAS community**.
> > >
> > > Thanks again for the time you spent on this insightful review!
> > >
> > > Best regards,
> > >
> > > Authors

---

### Official Review · Reviewer_LUTJ · 2025-03-11

**Overall Recommendation:** 4

**Summary:**

This paper introduces G-Designer, an innovative solution for multi-agent communication topology design in LLM-MAS. The authors first propose the Multi-agent Communication Protocol (MACP), which sets standards for LLM-MAS topology design in terms of effectiveness, complexity-adaptiveness, and adversarial robustness.
G-Designer models MAS as a graph, using a variational graph auto-encoder. It constructs a multi-agent network with a task-specific virtual node and then decodes an optimized communication topology. This is achieved through a process of encoding agents and task information, decoding a sketched graph, and refining it with anchor and sparsity regularization.
Extensive experiments on six benchmarks demonstrate G-Designer's superiority.

## update after rebuttal
The authors address most of my concerns. So I raised my score.

**Claims And Evidence:**

The core claim of this paper is that the topology of MAS should be task-dynamic. The authors first support this through the method design: G-Designer is trained to design a task-customized communication graph for different domains and user queries of varying difficulties. Secondly, experimental verification supports this: Fig 4 shows that different graphs for different queries can significantly reduce the average token consumption while maintaining good performance. Finally, case studies verify this: Fig 6 shows that G-Designer can indeed design different topologies for different tasks.

**Essential References Not Discussed:**

The authors have comprehensively discussed relevant papers.

**Experimental Designs Or Analyses:**

The benchmarks used in this paper are standard, and the authors comprehensively compared single-agent and multi-agent baselines. I have the following concerns:

 - In Section 5.1, why is the temperature set to 0 for single agents and 1 for other MAS? Could this introduce inconsistencies when reporting the performance?
 - Section 5.1 mentions that the number of dialogue rounds is K=3. Can the authors explain this? Would other values significantly affect performance? Can this value be automated rather than mannual defined?
 - The authors state that the anchor topology is set as a chain structure. Have the authors tried other structures?

**Methods And Evaluation Criteria:**

This paper is the first to use graph neural networks for topology design in LLM-MAS, which is novel. It uses the most commonly used datasets and evaluation metrics.

**Other Comments Or Suggestions:**

None

**Other Strengths And Weaknesses:**

### Strength

 - Using graph neural networks and variational autoencoding for adaptive MAS topology design seems quite novel.
 - The visualizations are all well-designed and clearly presented.
 - The experimental results of the method are good. Detailed ablation studies validate the method's advantages in performance, token efficiency, and adversarial robustness.

### Weakness

 - The authors only used GPT-3.5 and GPT-4. Have they considered using the latest GPT or LRMs?
 - Table 1 is limited to five agents. Have the authors considered testing with more agents (100 or 1000)? Can G-Designer maintain its cost efficiency with more agents?

**Questions For Authors:**

None

**Relation To Broader Scientific Literature:**

This paper is highly relevant to collaborative AI and multi-agent systems. It is the first to advocate for the task dynamics of LLM-MAS and proposes a unified protocol to regulate future LLM-MAS designs, which I believe is important for the community.

**Theoretical Claims:**

No new theorems or proofs provided.

---

> ### Author Rebuttal · Authors · 2025-03-31
>
> We sincerely thank you for the thoughtful and constructive reviews of our manuscript! Based on your questions and recommendations, we give point-by-point responses to your comments and describe the revisions we made to address them.
>
> ---
> > **`Weakness 1`**: Temperature Settings in the Experiments
>
> Thank you very much for your thorough review! In the multi-agent system, to facilitate more diverse communication among different agents, we followed DyLAN and set the temperature to 1. For the Single Agent scenario, to ensure more reproducible experimental results, we set the temperature to 0.
>
> For a fairer comparison, we have also presented the performance of the Single Agent method with the temperature set to 1 in `Table A`.  Experimental results show that the change in temperature from 0 to 1 does not significantly affect the average accuracy of single-agent methods.
>
> *Table A. Performance of single agent methods with temperature set to 1.*
> | Method | MMLU | GSM8K | HumanEval |
> |:-:|:-:|:-:|:-:|
> | Vanilla | 82.14 | 85.40 | 71.68 |
> | CoT | 82.55 | 86.81 | 75.80 |
> | ComplexCoT | 83.70 | 87.04 | 75.66 |
>
> ---
>
> > **`Weakness 2`**: The reason for setting the number of dialogue rounds.
>
> Your question is very valuable! In Table B, we present the changes in accuracy and cost with different numbers of dialogue rounds on the HumanEval dataset. Intuitively, increasing the number of optimization rounds leads to more refined and accurate results, yielding substantial performance improvements, but also comes with increased cost. To balance performance with token savings, we consistently set $K = 3$.
>
> *Table B. We report the performance of G-Designer on HumanEval benchmark with different $K$ values.*
> | K | 1 | 3 | 5 | 7 |
> |:-:|:-:|:-:|:-:|:-:|
> | Acc | 88.21 | 89.90 | 90.66 | 91.08 |
> | Cost | 0.1325 | 0.2298 | 0.3339 | 0.4097 |
>
> As regards the automation of $K$, we respectfully argue that G-Designer can easily achieve this via incorporating existing early-stopping mechanisms like in MacNet or DyLAN. Nevertheless, G-Designer not particularly leverage these, as its core contribution lies in the automation of MAS topology.
>
> ---
>
> > **`Weakness 3`**: The impact of different anchor settings on the results.
>
> The suggestion you proposed is very thoughtful! In Table C, we explored the impact of different initial topologies on the experimental results.
>
> *Table C: Different Topologies for anchor graph initialization on the GSM8K dataset.*
> | Anchor | Chain | Layered | Star | Random | FullConnected |
> | :-: | :-: | :-: | :-: | :-: | :-: |
> | Accuracy | 95.07 | 94.92 | 94.47 | 95.15 | 95.22 |
> | Token | $8.5\times 10^6$ | $9.3\times 10^6$ | $8.9\times 10^6$ | $9.2\times 10^6$ | $1.2\times 10^7$ |
>
> Using different topologies as initial anchors shows minimal performance differences, although the cost is higher. This aligns with our analysis in Section 5.4, where we concluded that the success of G-Designer is mainly attributed to its efficient communication graph generation.
>
> ---
>
> > **`Weakness 4`**: Have the authors considered using the latest GPT or LRMs?
>
> To address your concern, we use the newer and popular closed-source model GPT-4o-mini and the open-source model DeepSeek-V3 as base models, and compare the experimental results of our method with Vanilla and GPTSwarm in Table D.
>
> *Table D. Performance results with more advanced LLMs.*
> | Method | LLM | MMLU | GSM8K | HumanEval |
> |:-:|:-:|:-:|:-:|:-:|
> | Vanilla | GPT-4o-mini | 77.12 | 92.67 | 85.71 |
> | Vanilla | DeepSeek-V3 | 86.27 | 93.25 | 87.67 |
> | GPTSwarm | GPT-4o-mini | 78.43 | 93.78 | 86.28 |
> | GPTSwarm | DeepSeek-V3 | 86.93 | 94.38 | 89.72 |
> | G-Designer | GPT-4o-mini | 79.73 | 94.23 | 90.32 |
> | G-Designer | DeepSeek-V3 | 89.54 | 95.52 | 91.93 |
>
> Due to the time constraints of the rebuttal, we were unable to compare our method with more baselines. However, the experimental results presented above demonstrate that our method is also applicable to newer LLMs.
>
> ---
>
> > **`Weakness 5`**: Have the authors considered testing with more agents?
>
> Thank you for this thoughtful inquiry! In Table 6 of our paper, we provide a comparison of accuracy, time, token consumption, and cost across different agent number configurations. The experimental results show that G-Designer is still able to maintain its cost efficiency with more agents.

---

> > ### Comment · Reviewer_LUTJ · 2025-04-05
> >
> > Thank you for your thorough rebuttal. Your detailed responses and additional experiments in Tables A-D, effectively address my concerns about temperature settings, dialogue rounds, anchor topologies, and the use of newer LLMs. The results demonstrate the robustness and adaptability of G-Designer, reinforcing the paper’s overall solidity. Based on this, I’m happy to raise my score.

---

> > > ### Author Response · Authors · 2025-04-07
> > >
> > > **Dear Reviewer LUTJ**,
> > >
> > > We sincerely appreciate your invaluable support for our research. Your insightful suggestions regarding the anchor topology, temperature setting, and LLM backbone setting of G-Designer have significantly contributed to improving the depth and precision of our manuscript. We would also like to express our gratitude for your recognition of G-Designer's **novelty, clarity, robustness, and adaptability**.  It has been an honor to incorporate your comments and strengthen our work accordingly.
> > >
> > > Thank you once again for your time, expertise, and constructive review.
> > >
> > > Best regards,
> > >
> > > Authors

---

### Official Review · Reviewer_3iyh · 2025-03-15

**Overall Recommendation:** 3

**Summary:**

The paper discusses the advancements in collective intelligence among large language model-based agents, highlighting the challenge of selecting effective communication topologies for specific tasks. To address this, the authors introduce G-Designer, an adaptive solution that dynamically creates task-aware communication topologies using a variational graph auto-encoder. This approach aims to balance efficiency and performance by customizing the inter-agent communication network according to the task requirements.

**Claims And Evidence:**

Yes

**Essential References Not Discussed:**

All the essential references are discussed.

**Experimental Designs Or Analyses:**

Yes

**Methods And Evaluation Criteria:**

Yes

**Other Comments Or Suggestions:**

See above.

**Other Strengths And Weaknesses:**

Strengths:

1. It addresses an important problem of learning the communication network structure in a multi-agent system of LLMs.

2. Proposing a formal multiagent communication protocol is important. This paper attempts it.

3. Some analysis done in the paper w.r.t. the number of agents and number of tokens are very insightful.


Weaknesses / Confusions:

1.  In Section 3.2, what happened to the communication protocol if a graph has a directed cycle?

2. It is not clear why these 3 criteria are considered as the Mukti agent communication protocol? Is this list exhaustive?

3. The need of the simple starting anchor matrix \tilta{A} is not clear. If A is an almost random guess, why do you want your final topology to be close to it in Equation 14?

4. In Line 279, terms in Equation 14 are not correctly referred.

5. Equations 14 and 16 are both trainable optimisation functions with hyperparameters. Good amount of training data will be needed to optimize these objectives. The authors just mentioned that limited training data is required without explaining details.

6. The overall paper looks like an overkill and is often made complex without any proper motivation / intuition about where existing simpler things would fail?

7. In the implementation details, no details about the training of the proposed method is given. You need to provide training data name, size, validation and hyperparameter tuning details. Also, it is not clear if the baseline approaches need additional training. If yes, are they also being trained on the same datasets? If no, how do you ensure fairness in your experiments?

8. Can you clarify what all agents being used in Table 1? If they are all the same(GPT4 based), will all the nodes in the graph have similar node features and the final topology may boil down to a symmetric graph structure wrt the nodes?

9. The variance of G-Designer is high in Table 1? Please throw some light on the statistical significance of the results? Also, is it because of the additional uncertainties because of the training?



"**Update after rebuttal**"

I have checked the rebuttal. I thank the authors for their rebuttal. Some of my concerns are addressed. However, concerns around the correctness of the equations, motivation about some steps of the algorithm and details of the training are still there. I will hold my overall score.

**Questions For Authors:**

See above.

**Relation To Broader Scientific Literature:**

Yes, the contributions of the paper are related to the broader scientific literature of agentic frameworks of LLMs.

**Theoretical Claims:**

NA

---

> ### Author Rebuttal · Authors · 2025-03-31
>
> > **`Weakness 1`**: What happens to the communication protocol if a graph has a directed cycle?
>
> Thank you for the insightful inquiry! In Equation (15), we extract an enforced acyclic graph $\mathcal{G}\_{com}$ from the potentially cyclic topology $\tilde{\mathbf{S}}$. Specifically, in our provided anonymous code repository, the function `check_cycle()` in `GDesigner/graph/graph.py` is responsible for avoiding cycles when constructing $\mathcal{G}\_{com}$. We will emphasize this point more clearly in the revised version.
>
> ---
> > **`Weakness 2`**: Why are these 3 criteria considered?
>
>
> 1. **Effectiveness**: Extraordinary problem-solving is one of the core motivations behind MAS.
> 2. **Complexity Adaptiveness** aims to achieve optimal performance with minimal resource consumption. This is emphasized or pursued by many recent multi-agent papers, such as AgentPrune (ICLR 2025), RouterDC (NeurIPS 2024), and GraphRouter (ICLR 2025).
> 3. **Adversarial Robustness**: As attack & defense mechanisms in Agent/MAS gain increasing attention (e.g., TrustAgent (EMNLP 2024)), we believe safety and robustness will be critical for the trustworthy deployment of MAS in real-world applications.
>
> We respectfully acknowledge that while we have highlighted what we consider critical, we do not claim this framework to be exhaustive. We would greatly appreciate any additional insights you might suggest and would be happy to incorporate them into our future work.
>
> ---
> > **`Weakness 3`**: The starting anchor is simple.
>
> We would like to respectfully clarify that the anchor is not an "almost random guess". Instead, as stated in line 247, it incorporates prior knowledge of the workflow sequence (for example, a simple but standard coding procedure). In Section 5.4 and Table 3, we demonstrate that introducing the starting anchor results in a practical performance improvement.
>
> ---
> > **`Weakness 4`**: Equation 14 are not correctly referred.
>
> Thank you for your thorough review! We will correct the reference to Eq. (14) in Lines 278–288 in the revised version.
>
> ---
> > **`Weakness 5 & 7`**: No details about the training of the proposed method are given.
>
> - Training set: For both our method and the baselines that require training, we use the same training set. As mentioned in Line 321 (Right) of the paper: 40 samples for MMLU/MultiArith/HumanEval, 80 samples for GSM8K/SVAMP/AQuA;
> - Test set: provided in Table 4;
> - Hyperparameter tuning: We use a validation set of size 100 for each benchmark and apply a grid search to find the optimal parameters. To reduce the complexity of hyperparameter tuning for users, we provide a set of generalized hyperparameters in Section 5.1.
>
> ---
> > **`Weakness 6`**: The paper lacks proper motivation.
>
> Thank you for bringing this issue to our attention! We humbly provide a more intuitive explanation of each component:
>
> - Section 4.1: Makes it possible to process MAS with VGAE.
> - VGAE encoder $q(\cdot)$: Captures the semantic relationships among different agents regarding a given task.
> - VGAE decoder $p(\cdot)$: Directly leveraging complex MetaGPT/LLM-Debate/GPTSwarm will cause unnecessary costs on simple queries.
> - Equation 14: If the optimized topology is not regularized, it may suffer from malicious edges or unnecessary edges. This practice has been well-validated in the traditional graph structure learning literature, such as NeuralSparse (ICML 2020), PTDNet (WSDM 2021).
>
> ---
> > **`Weakness 7`**: If the baseline approaches need additional training?
>
> In the baselines of our paper, DyLAN, LLM-Blender, and GPTSwarm require training.
>
> We compare with other training-free methods because **all previously published training-required methods (including DyLAN (COLM'24), LLM-Blender (ACL'23) and GPTSwarm (ICML'24)) have compared with training-free methods**. Therefore, we follow the same setting as theirs for a comprehensive evaluation.
>
> Besides, we respectfully argue our approach has the following key advantages **in a fair way**:
>
> (1) G-Designer has **better performance** than training-free/required topologies; (2) its **training cost** is an order of magnitude lower than all training-required methods (Table 2). Additionally, our **inference cost** is among the lowest (Figure 4).
>
> ---
>
> > **`Weakness 8`**: Clarify the details of all the agents used in Table 1.
>
> The profiles and tools of different agents vary significantly, resulting in distinct encoded node features. Consequently, the constructed graph varies significantly. We provide a case study (Figure 6) to illustrate this. Examples of different agent role/tool descriptions can be found in our provided code in `GDesigner/prompt/gsm8k_prompt_set.py`.
>
> ---
> > **Weakness 9**: The variance of G-Designer is high in Table 1?
>
> We would like to clarify a possible misunderstanding: the subscripts in Table 1 indicate improvements over the vanilla baseline rather than variance.
>
> Besides, we would like to emphasize that all results in Table 1 are averaged over three runs.

---

### Decision · Program_Chairs · 2025-05-01

**Decision:**

Accept (spotlight poster)

**Comment:**

This work introduces G-Designer, an adaptive method that dynamically creates task-aware communication topologies using a variational graph auto-encoder. Through experiments, the authors show that the proposed method is promising in terms of performance, communication overhead, task-adaptive, and robustness.

The strengths of this work are summarized as follows.

1. It addresses an important problem of learning the communication network structure in a multi-agent system of LLMs.
2. The proposed method, G-Designer, can dynamically design task-aware, customized multi-agent communication topologies.
3. G-Designer is the first to use graph neural network and  variational autoencoder for topology design in LLM-MAS.
4. G-Designer is promising in terms of accuracy/pass@1 and communication tokens. Moreover, G-Designer is robust in face of adversarial attack.